# Effect of Aliphatic Aldehydes on Flavor Formation in Glutathione–Ribose Maillard Reactions

**DOI:** 10.3390/foods12010217

**Published:** 2023-01-03

**Authors:** Hao Liu, Lixin Ma, Jianan Chen, Feng Zhao, Xuhui Huang, Xiuping Dong, Beiwei Zhu, Lei Qin

**Affiliations:** 1National Engineering Research Center of Seafood, Collaborative Innovation Center of Seafood Deep Processing, School of Food Science and Technology, Dalian Polytechnic University, Dalian 116034, China; 2College of Food Science, Fujian Agriculture and Forestry University, Fuzhou 350002, China

**Keywords:** aliphatic aldehydes, ribose, glutathione, Maillard reaction, flavor, antioxidant activity

## Abstract

The Maillard reaction (MR) is affected by lipid oxidation, the intermediate products of which are key to understanding this process. Herein, nine aliphatic aldehyde–glutathione–ribose models were designed to explore the influence of lipid oxidation products with different structures on the MR. The browning degree, fluorescence degree, and antioxidant activity of the MR products were determined, and the generated volatile organic compounds (VOCs) and nonvolatile compounds were analyzed by gas chromatography-mass spectrometry and ultra-performance liquid chromatography-mass spectrometry. A total of 108 VOCs and 596 nonvolatile compounds were detected. The principal component and hierarchical clustering analyses showed that saturated aldehydes mainly affected the VOCs generated by the MR, while unsaturated aldehydes significantly affected the nonvolatile compounds, which changed the taste attributes of the MR products. Compared with the control group, the addition of unsaturated aldehydes significantly increased the sourness score and decreased the umami score. In addition, the addition of unsaturated aldehydes decreased the antioxidant activity and changed the composition of nonvolatile compounds, especially aryl thioethers and medium chain fatty acids, with a strong correlation with umami and sourness in the electronic tongue analysis (*p* < 0.05). The addition of aliphatic aldehydes reduces the ultraviolet absorption of the intermediate products of MR browning, whereas saturated aldehydes reduce the browning degree of the MR products. Therefore, the flavor components of processed foods based on the MR can be effectively modified by the addition of lipid oxidation products.

## 1. Introduction

The Maillard reaction (MR) and lipid oxidation are two of the most important chemical reactions in biochemical systems and play important roles in the development and imitation of flavor in processed foods. Nonenzymatic browning during the MR involves the condensation of reducing sugars with amino acids or peptides and typically occurs during food production and storage [1]. During this reaction, a series of complex compounds, called MR products (MRPs), are produced [2], which are usually regarded as natural flavors owing to their strong fragrance and natural reaction from raw materials. In recent years, studies have shown that MRPs extracted from food protein-derived peptides contribute not only to the flavor and color of food but also to the biological activity. Furthermore, these compounds possess strong antioxidant activity [3].

Lipids are also important components in food, and their degradation and oxidation can produce characteristic flavors. In particular, the typical aromas of matured or dry-cured meat products mainly originate from lipid oxidation [4]. The heat-induced oxidation of fatty acids, especially unsaturated fatty acids, produces degradation products, such as aliphatic aldehydes, ketones, and alcohols, which have intrinsic flavors [5].

Lipid degradation products directly interfere with aroma formation in processed foods through interactions with Maillard intermediates and lipid–Maillard interactions [6]. Both the MR and lipid oxidation reaction represent complete networks of different reactions, and the intermediate products interact directly or indirectly with each other, resulting in extremely complex mixtures of compounds. Many researchers have studied reaction systems based on amino compounds and reducing sugars to simulate the MR and explore how it is affected by lipid oxidation. Their results show that lipid–Maillard interactions are dependent on the reaction conditions, including the pH, temperature, and time [7]. Additionally, different types of lipids (phospholipids, triglycerides, and polyunsaturated fatty acids) have clear effects on the MR [8,9]. However, the structural characteristics of the lipid degradation products, such as the degree of (un)saturation and carbon chain length, may have a critical impact on the lipid–Maillard interaction mechanism; however, these characteristics are neglected by researchers. Furthermore, although the browning degree, volatile organic compound (VOC) formation, and antioxidant activity were evaluated as indicators of the MR in previous studies [10,11], nonvolatile compound formation may better explain the changes in taste.

Herein, nine aliphatic aldehydes, including three saturated aldehydes (hexanal, heptanal, and decanal), three monounsaturated aldehydes (*E*-2-hexenal, *E*-2-heptenal, and *E*-2-decenal), and three polyunsaturated aldehydes (*E*,*E*-2,4-hexadienal, *E*,*E*-2,4-heptadienal, and *E*,*E*-2,4-decadienal), were added to an aliphatic aldehyde–glutathione (GSH)–ribose system. Ultraviolet (UV) absorption and fluorescence analyses were performed to determine the content of the intermediate and end products of the MR. The odor and taste of the reaction solutions were determined using an electronic nose (E-nose) and electronic tongue (E-tongue), respectively. Changes in the profiles of the VOCs and nonvolatile compounds were analyzed by headspace solid-phase microextraction-gas chromatography-mass spectrometry (HS-SPME-GC-MS) and ultra-performance liquid chromatography-mass spectrometry (UPLC-MS), respectively. In addition, electron spin resonance (ESR) spectroscopy was performed to analyze the antioxidant activity of the reaction solutions.

## 2. Materials and Methods

### 2.1. Materials and Chemicals

Methanol and acetonitrile (HPLC gradient grade) were obtained from Spectrum Chemical Mfg. Corp. (Gardena, CA, USA). GSH (98%), ribose (>99.5%), hexanal (≥99%), *E*-2-hexenal (98%), *E*,*E*-2,4-hexadienal (>95%), heptanal (≥ 98%), *E*-2-heptenal (>95%), *E*,*E*-2,4-heptadienal (90%), decanal (97%), *E*-2-decenal (95%), and *E*,*E*-2,4-decadienal (>90%) were purchased from Aladdin Reagent Co., Ltd. (Shanghai, China).

### 2.2. Experimental Methods

#### 2.2.1. Model Reactions

For the preparation of MR model systems, GSH (0.30 mmol) and ribose (0.30 mmol) were dissolved in 5 mL of phosphate buffer (0.2 M), and the pH was adjusted to 6.5. Nine experimental groups were created by adding different aldehydes (0.03 mmol), as listed in Table 1. The samples were placed in an autoclave at 120 °C for 3 h for the reaction, then cooled immediately to room temperature. The experimental groups were named according to their compositions, as listed in Table 1.

#### 2.2.2. UV Absorbance and Fluorescence Analysis

According to the methods of Kim and Lee [12], the absorbance of the reaction solution was measured at wavelengths of 294 and 420 nm using a UV–visible absorption spectrometer (Lambda 35). The length of the light path of the colorimetric cell was 1 cm. The reaction solutions were diluted to the optimal measurement range (OD = 0.2–0.8) at different times as needed.

The fluorescence intensities of the reaction solutions were measured in the wavelength range of 200–700 nm [12]. A sample of the reaction solution (4 mL) was added to a quartz cuvette (10 × 18 × 45 mm) and placed in a fluorescence detector. To measure the MR degree, the excitation wavelength was set to 360 nm, and the emission wavelength was set to 400–600 nm. The scan rate was 12,000 nm/min, and the slit widths for both the excitation and emission wavelengths were 5 nm. Finally, the spectrograms were exported using FL Solutions software.

#### 2.2.3. E-Nose and E-Tongue Analysis

The E-nose analysis was performed using a commercial PEN3 E-nose (WinMuster Airsense Analytics, Inc., Schwerin, Germany). The E-nose contains ten metal-oxide semiconductor sensors with selectivity for different VOCs. The accompanying software (WinMuster) was used for data storage and multivariate statistical processing [13]. According to a previous method [14], the samples (500 μL) were placed in a 20 mL airtight vial and incubated for 30 min in a thermostatic water bath at 35 °C. Subsequently, the headspace gas was pumped into the sensor cavity at a constant rate of 100 mL/min through a Teflon tube attached to a needle, with clean air as the carrier gas. The volatile gas was delivered to the detector at a constant rate for up to 100 s to allow the sensor signal to reach a stable value. After the measurement was completed, a 500 s standby procedure was performed to clean the detector cavity with clean air until the sensor signals returned to the baseline range.

The taste attributes were analyzed using a TS-5000Z E-tongue (Insent Inc., Atsugishi, Japan) equipped with five sensors that evaluate salty, sour, umami, bitter, and astringent tastes. A reference solution and standard salty, sour, umami, bitter (+) and (−), and astringent solutions were prepared by dissolving the following compounds in 1 L of distilled water: reference solution, 0.045 g tartaric acid and 2.24 g KCl; salty solution, 0.045 g tartaric acid and 22.37 g KCl; sour solution, 0.45 g tartaric acid and 2.24 g KCl; umami solution, 0.045 g tartaric acid, 2.24 g KCl, and 1.69 g monosodium glutamate; bitter (+) solution, 0.045 g tartaric acid, 2.24 g KCl, and 0.04 g quinine hydrochloride; bitter (−) solution, 0.045 g tartaric acid, 2.24 g KCl, and 0.1 mL iso-α-acid; and astringent solution, 0.045 g tartaric acid, 2.24 g KCl, and 0.05 g tannic acid. The sample solutions were diluted with 90% (*v*/*v*) deionized water for the analysis. Prior to analysis, the sensors were calibrated and equilibrated. After the potentials of all membranes were stabilized in standard solutions, measurements were made for each sample. Finally, the experimental results were converted to taste values using TS-5000Z Management Server software.

#### 2.2.4. HS-SPME-GC-MS Analysis

The VOCs were separated using a nonpolar analytical column (HP-5MS Ultra Inert; 30 × 0.25 mm; 0.25 μm; Agilent Technologies) and identified by quadrupole mass spectrometry using an Agilent system (GC 7890B, MS 5977A MSD, Santa Clara, CA, USA). After minor adjustments according to a previous method [15], the analysis conditions were as follows: initial temperature of the chromatographic column, 35 °C for 3 min; heating rate, 15 °C/min; maximum temperature, 280 °C; final holding time, 5 min; injection method, split injection; injection temperature, 260 °C; carrier gas, helium at a flow rate of 1 mL/min. Mass spectra were obtained using a mass selective detector in the positive electron ionization mode at 70 eV in the scan range of 35–400. The ion source and transfer line temperatures were 230 and 280 °C, respectively. The compounds were semiquantified by comparing their mass spectra to the NIST 17 mass spectral library, and the content of each compound was calculated by comparing its area with that of the internal standards.

#### 2.2.5. UPLC-MS Analysis

The UPLC system was coupled to a Q-Exactive Orbitrap mass spectrometer (Thermo Fisher, CA, USA) equipped with a heated electrospray ionization probe. The peptides and amino acids in the samples were separated using an Acquity UPLC CSH-C18 column (2.1 × 100 mm; 1.7 μm; Waters Corporation, Milford, MA) coupled with a Waters Acquity BEH C18 VanGuard pre-column (2.1 × 5 mm; 1.7 μm; Waters Corporation) for identification. Mobile phases A and B were a formic acid–water solution (0.1%, *v*/*v*) and formic acid–acetonitrile solution (0.1%, *v*/*v*), respectively. The column temperature and flow rate were 40 °C and 0.4 mL/min, respectively. The elution gradient was as follows: 0 to 12 min, 5% to 10% B; 12 to 42 min, 10% to 35% B; 42 to 46 min, 35% B; 46 to 50 min, 35% to 50% B; 50 to 54 min, 50% to 95% B; 54 to 62 min, 95% B; 62 to 64 min, 95% to 5% B; 64 to 72 min, 5% B. The source parameters were as follows: spray voltage, 3.6 kV; capillary temperature, 380 °C; auxiliary gas heater temperature, 370 °C; sheath gas flow rate, 60 Arb; auxiliary gas flow rate, 25 Arb; purge gas flow rate, 2 Arb.

#### 2.2.6. ESR Analysis

**Determination of the hydroxyl radical clearance:** FeSO_4_·6H_2_O (189 mmoI) and ethylene diamine tetra-acetic acid (EDTA)-Na_2_ (0.586 mmoI) were dissolved in 1 L of deionized water, where the PH was adjusted to 7.4 with ammonia, to prepare an EDTA-Na_2_-Fe^2+^ solution. Phosphate buffered saline (PBS; 0.15 mol/L, pH 7.4) was prepared with deionized water as the solvent; then, 38 µL of PBS was placed into a 1.5 mL ep tube to which the sample solution (39 µL), radical scavenger (DMPO; 5 µL, 1 M), EDTA-Na_2_-Fe^2+^ (10 µL, 6 mM), and H_2_O_2_ (8 µL, 6% *v*/*v*) were added sequentially. The mixture was stirred and heated in a water bath at 40 °C for 30 min. Deionized water was used instead of the sample solution as a blank control, and the second peak was taken to represent the relative intensity of the signal.

**Determination of the DPPH solution clearance**: PBS (20 µL, 0.1 mol/L, pH 6.0), DPPH (20 µL, 0.2 mol/L), and the sample solution (10 µL) were added sequentially. Subsequently, the mixture was stirred and protected from light for 30 min to detect the reaction. PBS was used instead of the sample solution as a blank control, and the third peak was taken to represent the relative intensity of the signal.

#### 2.2.7. Statistical Analysis

All assays were performed in triplicate, and the results are expressed as the mean ± standard deviation (SD). Rank-dependent statistical processing was performed using SPSS 21.0 (SPSS Inc., Chicago, IL, USA). The statistical significance was set at *p* < 0.05. Untargeted data collected on a Q-Exactive Orbitrap mass spectrometer were analyzed using MS-DIAL 3.96. Partial least squares discriminant analysis (PLS-DA) was performed using MetaboAnalyst 4.0 [16].

## 3. Results and Discussion

### 3.1. UV Absorbance and Fluorescence Analysis

The preliminary experiments established that the MR model system was stable after reacting for 2.5 h at 120 °C and pH 6.5. The degree of browning, which is an important index for evaluating the MR, can be gauged effectively by measuring the UV absorbance at 294 and 420 nm [1,17]. The absorbance at 294 nm was indicative of the chromophores generated by the nonenzymatic browning during the intermediate stage of the MR, whereas that at 420 nm corresponded to the reaction products of the final stage [1]. Figure 1a,b show the absorbance of the reaction solution at 294 and 420 nm, respectively, with or without (control) different aliphatic aldehydes. When aldehydes were added to the MR solution, the UV absorbance at 294 nm was 34.6%–51.0% lower than that of the control group. As carbonyl compounds, aldehydes react with amino acids to generate pyridines when added to amino-acid-reducing sugar systems [18]. Aldehydes and ketones also react with the Strecker degradation products of amino acids to produce thiophenes and mercaptans [19,20], where the reaction products are related to the ratio between the aldehydes or ketones and amino acids [21] and is therefore affected directly by the exogenous addition of aldehydes. Thus, we believe that the aldehydes produced by the lipid oxidation changed the MR process by consuming the MR substrate (amino acids) and inhibiting the generation of the original MR intermediates. 

At 420 nm, the UV absorbance of the control group was higher than that of the groups with saturated aldehydes and lower than that of the groups with unsaturated aldehydes. The highest UV absorbance at 420 nm was measured for the G-R-VI-2 group (18% higher than that of the control group), while the lowest was measured for the G-R-VII-0 group (15% lower than that of the control group). In addition, the UV absorbance at 420 nm tended to have a positive correlation with the degree of unsaturation of the added aldehydes; that is, the absorbance was the highest for the polyunsaturated aliphatic aldehydes and lowest for the saturated aldehydes. Compared to saturated aliphatic aldehydes, unsaturated aliphatic aldehydes have higher activity and are therefore more likely to react with amino compounds in the form of carbonyl groups [9,22]. They are also prone to reverse acetal reactions, which generate two new carbonyl compounds [23] and promote the generation of melanin-like products. An exception to this trend was observed for *E*,*E*-2,4-decadienal (an unsaturated aliphatic aldehyde), which had a lower UV absorbance at 420 nm than that of *E*-2-decenal (the corresponding saturated aliphatic aldehyde). This demonstrates that unsaturated aldehydes with different carbon chain lengths can undergo different reverse acetal reactions and other reactions. As shown in Figure 1d, the reverse acetal reaction of *E*-2-decenal generated octanal and acetaldehyde. Meanwhile, the reverse acetal reaction of *E*,*E*-2,4-decadienal can generate 2-butenal and hexanal or 2-octenal and acetaldehyde [23]. Compared with the other unsaturated aldehydes, *E*,*E*-2,4-decadienal, which has a long carbon chain and high degree of unsaturation, is more unstable in the system and tends to inhibit melanin production during the MR.

The MR produces small molecules that generate fluorescence signals at 420–460 nm. These fluorescent substances are derived from small molecule intermediates produced by the degradation of Amadori compounds and Strecker degradation. The fluorescence spectra of the MR systems with different aldehydes are shown in Figure 1c. The addition of saturated aldehydes reduced the fluorescence intensity compared to that of the control group without aldehydes (G-R), while most of the unsaturated aldehydes enhanced the fluorescence intensity. The intensity of the fluorescence peak was positively correlated with the degree of unsaturation of the added aldehydes, which is consistent with the UV results in Figure 1b. The fluorescence peak intensity was the highest for the G-R-VI-2 group (25.6% increase compared with that of the control group) and the lowest for the G-R-VI-0 group (24.7% decrease compared with that of the control group). In addition, the fluorescence wavelength was red-shifted after adding aldehydes, wherein the degree of the shift was positively correlated with the degree of unsaturation of the aldehydes. Red-shifting of the florescence peak often occurs when chromophores (such as –OH, –NH_2_, and –NO_2_) are introduced into molecules to connect them to conjugated systems, which causes the maximum absorption wavelength to shift to a longer wavelength.

### 3.2. E-Nose and E-Tongue Analysis

Analyzing the generated VOCs and nonvolatile compounds is also important for evaluating the MR. E-nose and E-tongue systems provide a simple and effective means of measuring the changes in VOCs and nonvolatile compounds. An E-nose with ten sensors was employed to quickly identify changes in the contents of different types of compounds in the MR system. Compared to the control group, the systems with different aliphatic aldehydes altered the response of the E-nose sensors to varying degrees (Figure 2a–c). For most groups, the degree to which aliphatic aldehydes with the same carbon chain length influenced the E-nose sensor response was negatively correlated with the degree of unsaturation. The E-nose sensor with the most obvious change in response to the aldehyde-added systems was W5S, which reflects the nitrogen oxide content. The strongest signals detected by the W5S sensor were for the G-R-VII-0 and G-R-VI-0 groups (10.48- and 4.95-fold greater signal intensities, respectively, than that for the control group). In contrast, the signal detected by the W5S sensor for the G-R-X-0 group, despite containing a saturated aldehyde, had a slightly lower intensity (99.6%) than that of the control group. This demonstrates that the saturated aldehydes with different carbon chain lengths had significantly different influences on the MR. Notable changes were also recorded by the W1W (sulfur-containing compounds), W2W (sulfur-containing aromatic compounds), W1S (alkanes), and W2S (alcohols and some aromatic compounds) sensors. The results of the E-tongue analysis indicated that the aldehydes with different degrees of unsaturation changed the composition of the VOCs generated by the MR. To confirm this, we further analyzed the generated compounds by gas chromatography-mass spectrometry (GC-MS).

E-tongue systems use sensor arrays to simulate human taste buds to quickly detect flavor compounds in liquid samples with global selectivity [24]. The compounds detected by E-tongues correlate well with human sensory attributes [25,26] and can be used to effectively distinguish between samples with sensory evaluation. In addition, E-tongue systems can overcome the artificial subjectivity of sensory experiments and avoid potential safety risks for human testers; therefore, they are widely used [27].

The changes in taste between the control group and the aldehyde-added groups are shown in Figure 2d–f. Aftertaste A, aftertaste B, and richness represent the persistence of astringency, bitterness, and umami, respectively. The flavors in the aldehyde-added groups were mainly umami, astringency, sourness, and saltiness. The taste attribute that changed the most with the addition of aliphatic aldehydes was sourness. The sourness score of the control group (−13.04 ± 0.02) was the lowest, and its value gradually increased in the aldehyde-supplemented groups as the degree of unsaturation increased, with a smaller effect of carbon chain length. After adding saturated, monounsaturated, and polyunsaturated aldehydes, the sourness score increased by 4.67 ± 0.03, 7.98 ± 1.06, and 7.51 ± 1.08, respectively, compared with that of the control group. This may be due to the formation of acidic substances when the aldehydes were added to the MR system. The amino acids produced during the MR, such as cysteine, had a strong sour taste. Additionally, more CO_2_ was produced during the degradation process, and its dissolution in water increased the sourness. 

The umami score was the highest for the control (G-R) group (4.21 ± 0.02) and decreased by 1.37–3.04 for the aldehyde-added systems. Furthermore, the umami score was negatively correlated with the unsaturation of the aliphatic aldehydes. This phenomenon may be related to the degradation and metabolism of the peptides and amino acids. Glutathione can act as a source of glutamate compounds, which are associated with umami flavors [28]. In addition, the addition of aliphatic aldehydes enhanced the astringency and weakened the saltiness tastes, whereby the weakening of the saltiness was positively correlated with the degree of unsaturation of the aldehydes. However, there were no significant differences in bitterness between the groups. Compared with those of the other experimental groups, the aftertaste A and richness scores were significantly lower for the G-R-X-1 group, while the aftertaste B score increased. The results of the E-tongue analysis indicate that the addition of aldehydes promotes the consumption of some taste precursors and the generation of new taste compounds. To verify this, the nonvolatile compounds produced by the different systems were measured by UPLC-MS.

### 3.3. Differential Analysis of the MRPs Produced by MR Systems with Different Aliphatic Aldehydes Based on GC-MS and HPLC-MS

#### 3.3.1. Principal Component Analysis and Hierarchical Cluster Analysis

To further explore the influence of aliphatic aldehydes with different structures on the MR, GC-MS and UPLC-MS were used to determine the changes in VOCs and nonvolatile compounds. In total, 108 VOCs and 596 nonvolatile compounds were detected and identified.

Principal component analysis (PCA) and hierarchical cluster analysis (HCA) are unsupervised pattern recognition methods for classifying data based on the similarities and differences among the data. Figure 3a,b show the PCA score plots of the MRPs produced by the MR systems with different aliphatic aldehydes based on the VOC and nonvolatile compound data. As shown in Figure 3a, the first and second principal components (PC1 and PC2, respectively) of the PCA described a total of 35.5% (19.4% and 16.1%, respectively) of the variability in the GC-MS dataset. The PCA score plots demonstrate that the addition of saturated aldehydes had a clear effect on the volatile products of the MR; the saturated aldehyde groups were all distributed in the first quadrant of the score plot. These results are consistent with the E-nose results, whereby the responses of the sensors increased significantly for the groups with saturated aldehydes. Among the unsaturated aldehyde groups, the VOCs in the G-R-X groups also changed significantly and were mainly distributed in the second quadrant. As shown in Figure 3b, PC1 and PC2 of the PCA described a total of 57.8% (47.3% and 10.5%, respectively) of the variability in the UPLC-MS dataset. In contrast to the results obtained by GC-MS, the saturated fatty aldehyde groups and control group were similar in terms of nonvolatile compounds. Meanwhile, the addition of unsaturated fatty aldehydes caused the nonvolatile compounds in the MRPs to change significantly; the scores were mainly distributed in the fourth quadrant, except for those of the G-R-X-1 and G-R-VII-2 groups. According to the results of the E-tongue analysis, richness, aftertaste A, and aftertaste B changed most significantly in the G-R-X-1 group.

The dendrograms produced by the HCA are shown in Figure 3c,d. All of the samples were classified separately according to the changes in the generated VOCs and nonvolatile compounds. The results further prove the similarity between the the saturated aldehyde groups and the difference between the unsaturated aldehyde groups, whose double bond positions and carbon chain lengths were different. Compared with the control group, the addition of the saturated aldehydes significantly changed the VOCs generated during the MR, while the addition of the unsaturated aldehydes significantly changed the generated nonvolatile compounds.

Unsaturated aldehydes are less stable than saturated aldehydes and often undergo further oxidation to produce new VOCs and nonvolatile compounds [29]. In addition, unsaturated aldehydes undergo reverse acetal reactions [30]. The chain-breaking of saturated aldehydes by reverse acetal or reduction reactions can generate active intermediates that react with hydrogen sulfide or ammonia to generate new nitrogen- or sulfur-containing compounds. This complicates the mechanism by which aliphatic aldehydes influence the MR system. The degree of unsaturation directly affects the reactivity of aldehydes [31], and its position also affects the composition of the generated products [32]. The carbon chain length of unsaturated aldehydes has also been shown to affect their reactivity and oxidation products [33].

#### 3.3.2. VOC Analysis Based on GC-MS

In addition to the externally added aliphatic aldehydes, the MR itself also produces a certain number of intermediate aldehydes owing to Strecker degradation, which directly or indirectly affect the subsequent reactions [34]. As shown in Appendix A, a certain amount of heptanal and *E*,*E*-2,4-heptadienal were produced in the G-R group. Comparing the G-R-VI-0 group with the G-R-VI-1 group, it was found that the response of hexanal in the latter system was three-fold higher than that in the former, which indicates that some hexenal was converted to hexanal. G-R-VII-2 had 184.3-fold increased *E*-2-heptenal, G-R-X-1 had 42.8-fold increased heptanal, and G-R-X-2 had 8.8-fold increased *E,E*-2,4-heptadienal. This phenomenon may be related to the reverse acetal reactions of unsaturated aldehydes, which would generate many chain-breaking aldehydes with low unsaturation [30].

Strecker degradation is the main reaction pathway to produce aliphatic aldehydes during the MR. The aldehydes produced by this pathway are called Strecker aldehydes. In addition, α-aminocarbonyl compounds are formed after a series of processes, including Schiff base rearrangement, decarboxylation, and hydrolysis. These compounds can act as precursors for the generation of flavor substances, such as furans and pyrans [6]. Furans are nitrogen-containing heterocyclic compounds; they are typical carcinogens and tend to have low flavor thresholds. Two pathways have been demonstrated for their formation in the presence of amino acids such as GSH; the first is an amino acid-dependent pathway that occurs through the recombination of reactive C_2_ and/or C_3_ fragments, and the second occurs by derivation of the complete sugar backbone independently of but promoted by amino acids (most amino acids, including cysteine, promote this pathway) [35]. Appendix A shows that 2-*n*-butyl furan had the highest response among the furans in the G-R group; however, its response was 62.2% lower in the G-R-X-0 group. Additionally, 1-(5-methyl-2-furanyl)-1-propanone and 5-methyl-furfural showed great responses in the G-R-VII-0 and G-R-X-2 groups, respectively; their responses were 334.6- and 676.4-fold higher than those of the control group, respectively. The production pathways of furans are complex and related to the formation of pigments in the later stages of the MR [35].

Six thiophenes were detected in the GSH–ribose MR systems, which were mainly generated by the exchange of sulfur and oxygen. This process occurs on hydrogen sulfide and furan or furanone substrates, which are generated by the degradation of GSH [36]. The detected thiophenes included α-methyl-thiophene, 3-thiophenecarboxaldehyde, and propyl-thiophane, the responses of which all increased with the addition of aliphatic aldehydes.

In the G-R-X group, many new VOCs—mainly thiazoles and pyridines—were produced. In the G-R-X-0, G-R-X-1, and G-R-X-2 groups, the responses of 3-(2-pyridyl) propyl acetate were 8.45-, 28.82-, and 14.95-fold higher, respectively, than that in the control group. The addition of decadienal also produced a new thiazole, 2-butyl-4-ethyl-5-methylthiazole, owing to the competition between the two carbonyl compounds produced by the MR and the changed reaction pathway. In the G-R-X-2 group, the response of 2-butyl-4-ethyl-5-methylthiazole was 363.5-fold greater than that in the control group. In the G-R-X groups, the responses of ketones and esters also increased to varying degrees, whereas their responses were lower in all of the G-R-VI and G-R-VI groups. Therefore, aldehydes with long carbon chains can effectively change the MR pathways and increase the variety of products, thereby generating new flavors.

#### 3.3.3. Nonvolatile Compound Analysis based on UPLC-MS

To further explain the taste attributes detected by the E-tongue, a correlation analysis was performed between the detected taste attributes and the taste compounds identified by UPLC-MS, as shown in Figure 4. Among the nonvolatile compounds identified by UPLC-MS, 47 flavor compounds were screened and divided into 14 categories. The E-tongue sensors comprise polymer membranes doped with artificial lipids [37]. When taste compounds contact the sensors, the potential of the membrane changes as the compounds combine with the lipid-like molecules. The E-tongue uses these changes in the membrane potential to detect the taste attributes [38]. Almost all of the selected taste compounds had a weak negative correlation with astringency. This may be due to the interaction between these compounds and the lipid-like membranes of the E-tongue sensors. With the exception of aryl thioethers, the contents of the other flavor compounds were all positively correlated with the degree of bitterness. The aryl thioethers were positively correlated with saltiness and umami, but negatively correlated with sourness and aftertaste B (*p* ≤ 0.05). Methoxyphenols, amino acids, and ketones exhibited the same trend. However, penams, dicarboxylic acids, and medium chain fatty acids had the opposite effects to aryl thioethers. In particular, the medium chain fatty acids had the most significant effect, and penams were positively correlated with richness (*p* ≤ 0.05). Owing to their similar hydrophilic/lipophilic balance, medium chain fatty acids destroy the stability of the cell membrane bilayer in the form of itself incorporation of lipid bilayers [39] and change the pore structure of the membrane [40], which affects the signals detected by the E-tongue sensors. The other flavor compounds had no obvious correlation with umami but were slightly positively correlated with sourness and saltiness.

### 3.4. Antioxidant Activity Analysis

MRPs, represented by melanoidins, have significant antioxidant effects in vitro, in vivo, and in food systems. They are mainly derived from the formation of Amadori products by pyrolysis in the primary stage of the MR [41]. In addition, heterocyclic products derived from the MR or sugar caramelization may also affect the production pathways of MRPs [42].

Determining the DPPH and hydroxyl radicals is a common method of evaluating the in vitro antioxidant activities of MRPs. Previous studies have shown that the UV spectra are affected by the absorption of MR browning products [43] and other free radicals [44]. Unique to free radicals, ESR spectroscopy has been proven to be a more selective and reliable tool for measuring the antioxidant capacity, and the ESR results correlate highly to UV results [45]. By comparing the antioxidant capacities of the different MRPs, their specificities and similarities were compared. Hydroxyl radicals are nonselective oxidants and considered the most reactive type of oxygen-containing radical [46,47]. Figure 5a–c show that the control group had a better scavenging effect on hydroxyl radicals than the aldehyde-added groups. Melanoidin and oxygen-containing and/or nitrogen-containing heterocyclic compounds, such as furan and pyrazine, were generated during the MR. These compounds contain an uneven distribution of π electrons and excess electrons on the carbon atoms, which increase the π electron cloud density and promote electrophilic addition, thus greatly increasing the ability of the compound to scavenge free radicals [48]. Therefore, it is speculated that the unsaturated aldehydes have high reactivity and form intermediates with GSH, which inhibits the progress of the MR. This result agrees well with the E-nose results for the G-R-VI and G-R-VII groups. The responses of the nitrogen- and oxygen-containing compounds in the systems containing hexanal and heptanal were much higher than those in the systems containing unsaturated aldehydes; however, this result was not observed for the G-R-X groups. This may be because the carbon chain of decanal is longer than those of hexanal and heptanal, and the reaction mechanism is more complicated, although the exact mechanism is still unclear.

DPPH is a purple compound containing chromogen-free radicals that can form stable yellow DPPH-H molecules under the action of antioxidants. Studies have shown that MRPs can scavenge DPPH [3]; therefore, this scavenging reaction was used to determine the effect of aliphatic aldehydes on the antioxidant activity of the MRPs. Figure 5d–f show that the control group had the most obvious scavenging effect on hydroxyl radicals, followed by that of the saturated aldehyde groups. For all of the aldehyde-added groups, the DPPH scavenging effect was negatively correlated with the degree of unsaturation of the aldehydes. In addition, the unsaturated aldehyde groups had a poor scavenging effect on the hydroxyl radicals. This may be because aliphatic aldehydes with more unsaturated bonds have stronger activity and are therefore more likely to react with GSH and inhibit the MR. As a result, the amount of melanoidins produced was small, thus weakening the free radical scavenging effect and antioxidant capacity. This result also confirms the conclusion of the E-nose and hydroxyl radical experiments.

## 4. Conclusions

In this study, the effects of aldehydes with different structures on the MR were analyzed in terms of the browning degree, generated VOCs and nonvolatile compounds, and antioxidant activity. Saturated aldehydes reduced the browning degree of the MRPs and increased the responses of the W5S, W1W, and W2W sensors of the E-nose. The fluorescence intensity of the MRPs increased with the degree of unsaturation of the added aldehydes; unsaturated aldehydes increased the sourness score of the MRPs, as detected by the E-tongue, by 7.75 ± 1.32. Based on 108 VOCs measured by GC-MS and 596 nonvolatile compounds measured by UPLC-MS, the PCA and HCA results showed that saturated aldehydes mainly changed the types and contents of VOCs in the MRPs, whereas unsaturated aldehydes significantly affected the nonvolatile compounds in the MRPs. The addition of aldehydes reduced the antioxidant activity of the MRPs, with the inhibitory effect of unsaturated aldehydes being greater. By analyzing several MR models, the effects of different saturated and unsaturated aliphatic aldehydes on the MR were assessed in this study. The results advance our understanding of lipid–Maillard interactions and lay a solid foundation for the regulation and development of flavors in MR-processed foods in the future.

## Figures and Tables

**Figure 1 foods-12-00217-f001:**
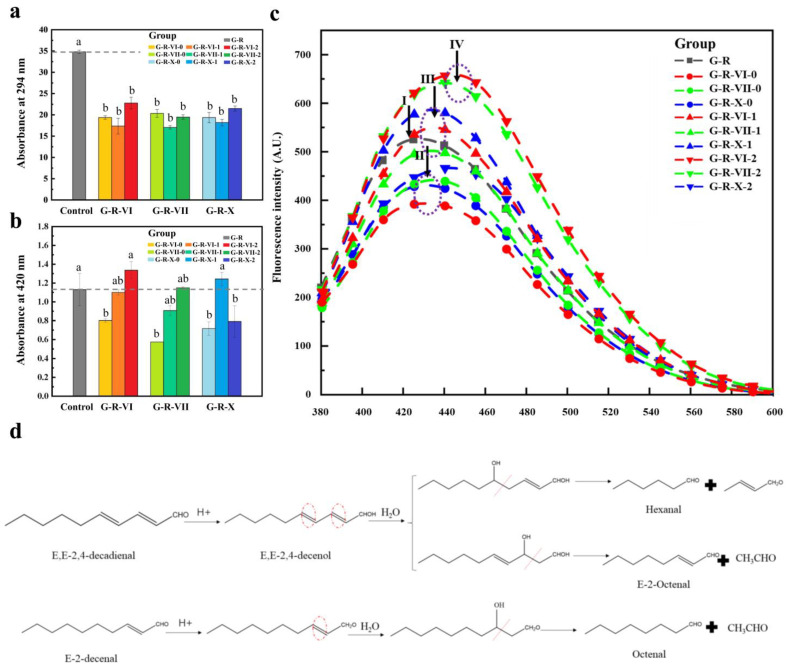
UV absorbance and fluorescence analysis of MRPs produced by MR systems with different aliphatic aldehydes: (**a**) absorbance at 294 nm, (**b**) absorbance at 420 nm, (**c**) fluorescence spectra; (**d**) reverse acetal reaction pathways of *E*,*E*-2,4-decadienal and *E*-2-decenal. Different lowercase letters denote significant differences (*p* < 0.05).

**Figure 2 foods-12-00217-f002:**
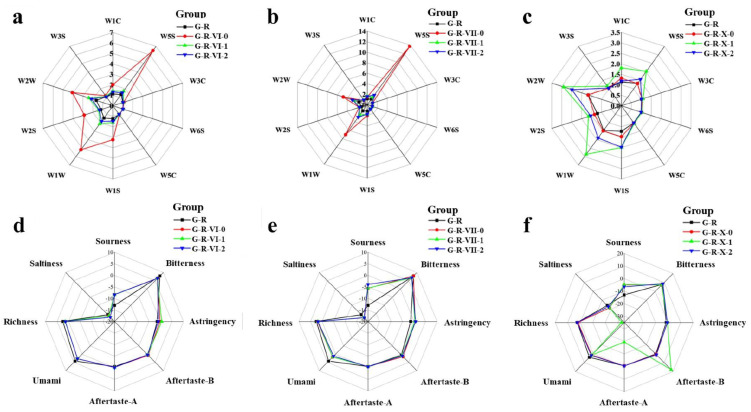
E-nose and E-tongue analysis of MRPs produced by MR systems with different aliphatic aldehydes: (**a**–**c**) E-nose and (**d**–**f**) E-tongue analysis of the (**a**,**d**) control (G-R) and G-R-VI groups (G-R-VI-0, G-R-VI-1, and G-R-VI-2), (**b**,**e**) G-R and G-R-VII groups (G-R-VII-0, G-R-VII-1, and G-R-VII-2), and (**c**,**f**) G-R and G-R-X groups (G-R-X-0, G-R-X-1, and G-R-X-2).

**Figure 3 foods-12-00217-f003:**
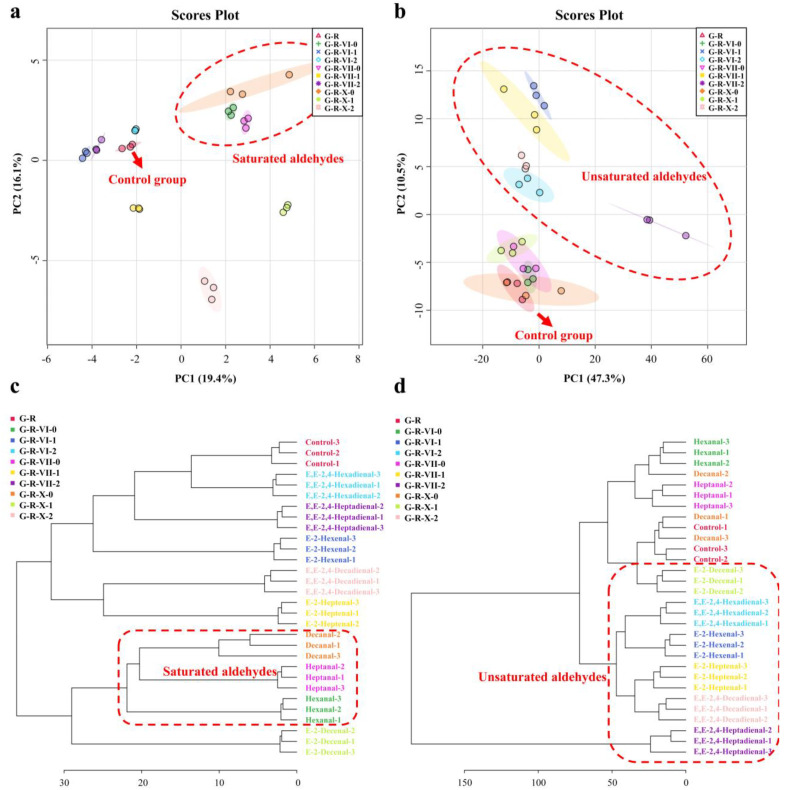
PCA score plots and HCA of the MRPs produced by the MR systems with different aliphatic aldehydes. Sample clustering based on GC-MS and UPLC-MS results by (**a**,**b**) PCA and (**c**,**d**) HCA.

**Figure 4 foods-12-00217-f004:**
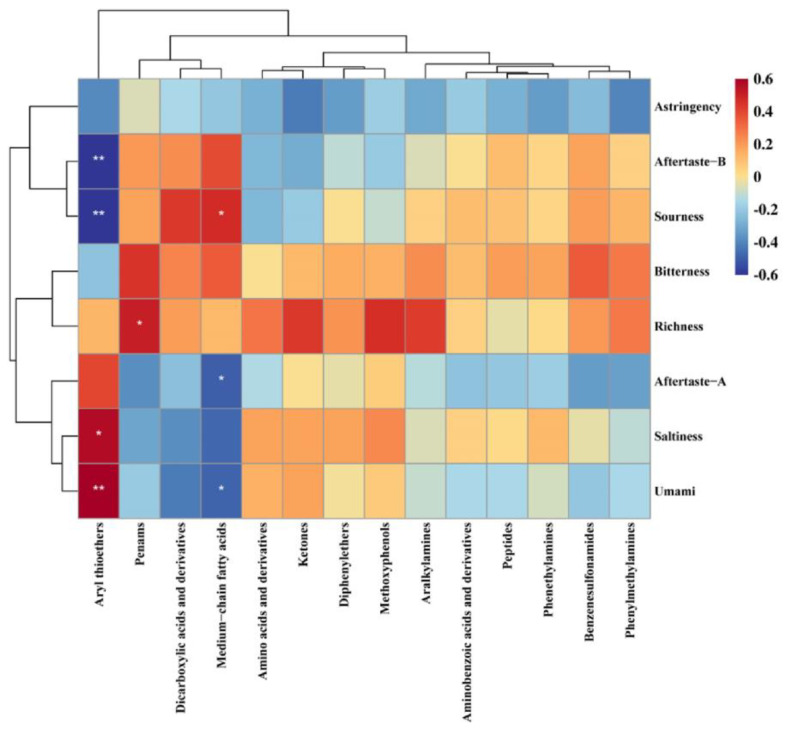
Correlation heatmap between the taste attributes and the content of nonvolatile taste compounds. * *p* < 0.05; ** *p* < 0.01.

**Figure 5 foods-12-00217-f005:**
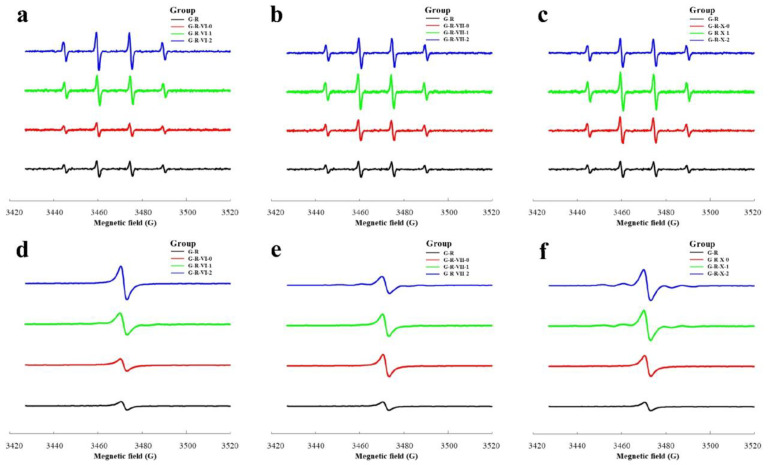
ESR spectra of the MRPs produced by the MR systems with different aliphatic aldehydes: (**a**–**c**) hydroxyl radicals and (**d**–**f**) DPPH of the (**a**,**d**) control (G-R) and the G-R-VI groups, (**b**,**e**) G-R and G-R-VII groups, and (**c**,**f**) G-R and G-R-X groups.

**Table 1 foods-12-00217-t001:** Composition and notation of the reaction systems in each experimental group.

Abbreviation	Class	Components		
		Amino Compounds	Reducing Sugars	Aliphatic Aldehydes
G-R	Control group	GSH	ribose	-
G-R-VI-0	G-R-VI groups	GSH	ribose	Hexanal
G-R-VI-1		GSH	ribose	*E*-2-hexenal
G-R-VI-2		GSH	ribose	*E*,*E*-2,4-hexadienal
G-R-VII-0	G-R-VII groups	GSH	ribose	Heptanal
G-R-VII-1		GSH	ribose	*E*-2-heptenal
G-R-VII-2		GSH	ribose	*E*,*E*-2,4-heptadienal
G-R-X-0	G-R-X groups	GSH	ribose	Decanal
G-R-X-1		GSH	ribose	*E*-2-decenal
G-R-X-2		GSH	ribose	*E*,*E*-2,4-decadienal

## Data Availability

The datasets generated for this study are available upon request to the corresponding author.

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
