# Peer review of "Effect of Aliphatic Aldehydes on Flavor Formation in Glutathione–Ribose Maillard Reactions"

_foods, 2023, doi:10.3390/foods12010217_

Round 1

Reviewer 1 Report

The manuscript entitled „Effect of Fatty Aldehydes on Flavor Formation in Glutathione-Ribose Maillard Reaction” describes a study with a model of Maillard reaction with added different aldehydes.

In my opinion, the manuscript does not meet the quality criteria to be published in this form.

The text contains many errors, is poorly written and is not concise. One sentence has no connection with another which is the most visible in the abstract, which is bad.

The authors use some strange nomenclature such as “fatty aldehydes”, “volatile components”, and “sulfocompounds” What is that? None of this is a correct name.

The introduction is not clear. I do not understand the meaning of lines 61-69. What is this paragraph about and what is the importance of the article?

Based on lines 70-80, it looks like this study has no novelty. I do not see a clear presentation of the research gap and novelty.

The authors should not repeatedly report the analysis of absorbance at 294 and 420 nm. What does it mean? It is only a tool to measure something, so the focus should be put on the measured parameter, not on the wavelength.

Why the analysis was performed with ESR? DPPH can be measured with simple UV. Moreover, DPPH is not the best method, which is not very selective and reliable.

The worst and most disqualifying part of this manuscript is the analysis of VOCs. Firstly, when only the single quad is used, there is no chance for identification. Other solutions are needed. Moreover, what is this “semi-quantification”? In response to a single internal standard?? It is just wrong. Please read the recent commentary which explains well, that such an approach is wrong, bad and cannot be ever used! (https://doi.org/10.1016/j.jfca.2022.104955). Therefore all the conclusions and discussion of VOCs are incorrect in this manuscript since is based on biased results. And these are volatile organic compounds, not components!

Line 203-204 – but how?

Line 212 – participate how?

Line 233 – provide the correlation results.

Line 424-434 – why is it in section 3.4, not 3.5, which is about UHPLC?

Why the PLS analysis was performed only on UHPLC data, not on all? To prove what? I do not understand this cherry-picking approach.

Reviewer 2 Report

Review of:  Effect of Fatty Aldehydes on Flavor Formation in Glutathione-Ribose Maillard Reactions.

General comments:  This manuscript examines compounds produced on heating model systems containing various fatty aldehydes with glutathione (a small peptide) and a sugar source.  These reactions are known to produce various Maillard compounds that are associated with cooked food flavors.  This study provides needed information as to the mechanism of such reactions.

Title:  Suggest changing title to “…Maillard Reactions.”

Line 28:  Remove “obviously” 

Line 64:  Change “El-massry” to “El-Massry”

Lines 73 to 74:  Reword to read “...proven that have effects on MR [15,16].  Hexanal, E-2-heptenal, and E, E-2,4-decadienal were found to be the main reactive species…”

Lines 76-77:  Reword to read “…is helpful for understanding the lipid-Maillard interaction and to find the changes of flavor components.”

Line 79:  Remove “huge”

Lines 81 and 82:  Change “aldehyde” to “aldehydes”

Line 106:  What is meant by “a high-pressure sterilizer”?  Do you mean an autoclave?

Lines 108 to 109:  Change to read “…in each experimental group, the abbreviations were set as shown in Table 1, below.”

Paragraph beginning with line 138:  What are the compositions of the solutions used to calibrate the sensors?  More information is needed as to how this was done.  How does it correlate with actual human tastes?

Line 318:  Reword to read “… of the total, but is significantly…”

Line 322:  Remove “And” and start the sentence with “Their contents…”

Line 357:  Remove “And” and start the sentence with “They were…”

Line 368:  Add “an” before “uneven”

Linde 369:  Add “the” before “carbon atoms”

Lines 414 to 416:  Monosodium glutamate is the salt form of glutamic acid.  This statement is does not really make sense.  Simply state that glutamate compounds are associate with umami flavors.

Paragraph beginning with line 424:  Why would these compounds inhibit astringency?  Are they blocking prolines?  A justification with a reference is needed here.

Line 469:  Change to read “A VIP map is presented at Figure 5.”

Line 524:  Change “fat” to “fatty”

Lines 527 to 528:  Reword to read “…the umami flavor intensity decreased to 2.56 ± 0.48.  The addition of aldehydes inhibited the antioxidant activity of MR, and the inhibiting effect of the unsaturated aldehydes was greater.”

Reference format:  Please read the instructions to authors for the proper format.  The format here is not consistent and is often wrong.  Do not capitalize the titles except for the first word and proper nouns.  Abbreviate all the journal names.
